# Evaluation of association studies and a systematic review and meta-analysis of *CYP1A1* T3801C and A2455G polymorphisms in breast cancer risk

Chen Yang[1], Xiao-Feng He[2]*

1 Teaching Reform Class of 2016 of the First Clinical College, Changzhi Medical College, Shanxi, Changzhi, China, 2 Institute of Evidence-based medicine, Heping Hospital Affiliated to Changzhi Medical College, Shanxi, Changzhi, China

* 393120823@qq.com

**Data Availability Statement:** All relevant data are within the paper and its Supporting Information files.

## Abstract

### Background

Nine previous meta-analyses have been published to analyze the *CYP1A1* T3801C and A2455G polymorphisms with BC risk. However, they did not assess the credibility of statistically significant associations. In addition, many new studies have been reported on the above themes. Hence, we conducted an updated systematic review and meta-analysis to further explore the above issues.

### Objectives

To explore the association on the *CYP1A1* T3801C and A2455G polymorphisms with BC risk.

### Methods

Preferred Reporting Items for Systematic Reviews and Meta-Analyses (The PRISMA) were used.

### Results

In this study, there were 63 case–control studies from 56 publications on the *CYP1A1* T3801C polymorphism (including 20,825 BC cases and 25,495 controls) and 51 case–control studies from 46 publications on the *CYP1A1* A2455G polymorphism (including 20,124 BC cases and 29,183 controls). Overall, the *CYP1A1* T3801C polymorphism was significantly increased BC risk in overall analysis, especially in Asians and Indians; the *CYP1A1* A2455G polymorphism was associated with BC risk in overall analysis, Indians, and postmenopausal women. However, when we used BFDP correction, associations remained significant only in Indians (CC *vs*. TT + TC: BFDP < 0.001) for the *CYP1A1* T3801C polymorphism with BC risk, but not in the *CYP1A1* A2455G polymorphism. In addition, when we further performed sensitivity analysis, no significant association in overall analysis

**Funding:** The authors received no specific funding for this work.

**Competing interests:** The authors have declared that no competing interests exist.

and any subgroup. Moreover, we found that all studies from Indians was low quality. Therefore, the results may be not credible.

## Conclusion

This meta-analysis strongly indicates that there is no significant association between the *CYP1A1* T3801C and A2455G polymorphisms and BC risk. The increased BC risk may most likely on account of false-positive results.

## Introduction

Breast cancer (BC) is one of the most common cancers and the main cause of cancer mortality among women worldwide. Moreover, the incidence rate of BC is unequal in different areas and races [1, 2]. Cumulative evidence indicated that environment, lifestyle, tobacco, alcohol consumption, gene, and several reproductive factors were important risk factors for BC [3–6]. In recent years, the study on gene polymorphism has received much attention in the development of BC worldwide [7, 8].

Cytochrome *P450 1A1* (*CYP1A1*), which codes the enzyme cytochrome *P450 1A1*, is a pivotal gene in metabolism of carcinogens, particularly polycyclic aromatic hydrocarbons (PAHs) [9–11]. PAH gain carcinogenicity once they are activated by xenobiotic-metabolizing enzymes into highly reactive metabolites [12]. Phase-I metabolic reaction is catalyzed by Cytochrome *P450* enzyme, and *CYP1A1* was considered to be the most foremost enzyme which catalyzes these PAHs to highly reactive metabolites [13]. Therefore, *CYP1A1* plays an important role in the etiology of BC. *CYP1A1* T3801C and A2455G are two of the common polymorphisms and they have been explored on their potential impacts with risk of BC. Hence, potential roles of *CYP1A1* polymorphisms with BC risk have been assumed [14, 15].

Both candidate-gene based and genome-wide association studies (GWAS) have revealed several significant loci associated with breast cancer in different cancer-regulating pathways [16–18] that modify the risk toward breast carcinogenesis. However, the genetic association studies subcontinent are primarily candidate association studies and have often reported contradictory results. Moreover, in the past decade, nine meta-analyses have been published to investigate the association between the *CYP1A1* T3801C and A2455G polymorphisms and BC risk [19–27]. However, the results of these meta-analyses were also contradictory and heterogeneous (S1 Table). Finally, 88 studies [S1 Appendix References] have been reported to evaluate the association between the *CYP1A1* T3801C and A2455G polymorphisms and risk of BC in different populations. However, results were still contradictory. Hence, we performed an updated systematic review and meta-analysis to assess the association on the above two issues.

## Materials and methods

The current systematic review and meta-analysis were conducted according to the Preferred Reporting Items for Systematic Reviews and Meta-Analyses guideline [28].

### Search strategy

A systematic literature search was conducted using the PubMed, Scopus, Embase, Chinese Biomedical Medical databases (CBM), China National Knowledge Infrastructure (CNKI), and WanFang databases (update to 15 July, 2020) by the following search strategy: (*CYP1A1* OR

cytochrome *P-450* OR cytochrome *P450*) AND (polymorphism OR variant OR variation OR mutation OR SNP OR genome-wide association study OR genetic association study OR genotype OR allele) AND breast. No language restriction was applied in the eligible studies. Additional studies have been screened out from the references of reviews and meta-analyses that published in the past decade. All the eligible studies were identified by reading the title, abstract, and full text of literatures. Moreover, we contacted the corresponding authors to obtain detailed information by e-mail if necessary.

## Inclusion and exclusion criteria

Eligible studies were included if they met the following criteria: (1) studies must be based on case-control or cohort studies; (2) genotype frequencies or odds ratios (ORs) and 95% confidence intervals (CIs) must be provided; (3) studies must investigate the association between the *CYP1A1* T3801C and A2455G polymorphisms and risk of BC. Exclusion criteria were as listed below: (1) articles were not on BC, (2) studies didn't provide the genotype data or ORs and 95% CIs, (3) for multiple publications of the same data, we only included the data from the largest or the latest studies.

## Data extraction and quality assessment

Data extraction and quality score assessment were performed by two authors (Yang and He) using pre-designed tables independently and was cross-checked for consensus to ensure its accuracy. Conflicts were discussed between the two authors to reach an agreement. The following information was collected from each study: first author, year of publication, country, ethnicity, source of controls, sample size, genotype distribution for cases and controls, and matching.

Quality assessment was performed by the two authors independently with a pre-designed scoring scale by one previous meta-analysis [29] (Table 1). The total score ranged from 0 to 20. Studies with scores 0–7, 8–13, or 14–20 were of low, moderate, or high-quality by two previously published meta-analyses [30, 31], respectively.

## Statistical analysis

Crude ORs and 95% CIs were used to estimate the association between the *CYP1A1* T3801C and A2455G polymorphisms and the risk of BC. The *CYP1A1* T3801C and A2455G polymorphisms were analyzed using the following five genetic models: CC vs. TT/GG vs. AA, TC vs. TT/AG vs. AA, CC vs. TC + TT/GG vs. AG + AA, CC + TC vs. TT/GG + AG vs. AA, and C vs. T/G vs. A.

We used $Q$ test and $I^2$ value to check heterogeneity among between-study heterogeneity (significant heterogeneity was regarded if $P < 0.01$ and/or $I^2 > 50\%$) [32]. For each genetic model contrast, summary ORs were calculated using random-effects model [33, 34]. The random-effects model was applied by the following two main reasons: (1) because the $Q$ test is characterized by low statistical power for between-study heterogeneity, which is especially relevant when few studies are available; (2) Usually, the random-effects model is a more conservative choice when heterogeneity is present, whereas it reduces to the fixed effect model when heterogeneity is absent. Subgroup analyses were calculated to assess the effects in the Asians, Caucasian, African, and Indian. Further subgroup analysis was conducted by menopausal status. Moreover, a meta-regression analysis was applied to explore the source of heterogeneity. Furthermore, a sensitivity analysis was performed by the following methods: a single study was removed each time and a dataset was used that the comprised only high-quality studies, matching studies, HWE, and genotyping performed blindly or with quality control [35]. Chi-

**Table 1. Scale for quality assessment of molecular association studies of BC.**

| Criterion | Score |
|---|---|
| Source of case | |
| Selected from population or cancer registry | 3 |
| Selected from hospital | 2 |
| Selected from pathology archives, but without description | 1 |
| Not described | 0 |
| Source of control | |
| Population-based | 3 |
| Blood donors or volunteers | 2 |
| Hospital-based | 1 |
| Not described | 0 |
| Ascertainment of cancer | |
| Histological or pathological confirmation | 2 |
| Diagnosis of BC by patient medical record | 1 |
| Not described | 0 |
| Ascertainment of control | |
| Controls were tested to screen out BC | 2 |
| Controls were subjects who did not report BC, no objective testing | 1 |
| Not described | 0 |
| Matching | |
| Controls matched with cases by age | 1 |
| Not matched or not described | 0 |
| Genotyping examination | |
| Genotyping done blindly and quality control | 2 |
| Only genotyping done blindly or quality control | 1 |
| Unblinded and without quality control | 0 |
| Specimens used for determining genotypes | |
| Blood cells or normal tissues | 1 |
| Tumor tissues or exfoliated cells of tissue | 0 |
| HWE | |
| HWE in the control group | 1 |
| HWD in the control group | 0 |
| Association assessment | |
| Assess association between genotypes and BC with appropriate statistics and adjustment for confounders | 2 |
| Assess association between genotypes and BC with appropriate statistics without adjustment for confounders | 1 |
| Inappropriate statistics used | 0 |
| Total sample size | |
| >1000 | 3 |
| 500–1000 | 2 |
| 200–500 | 1 |
| <200 | 0 |

HWE: Hardy-Weinberg equilibrium; HWD: Hardy-Weinberg disequilibrium; BC: breast cancer.

square goodness-of-fit test was used to check Hardy-Weinberg equilibrium (HWE), and statistically significant deviation was considered in control groups if $P < 0.05$ [36]. In addition, a Bayesian false discovery probability (BFDP) was used to correct multiple comparisons [37]. A cutoff value of BRDP was set up to be a level of 0.8 and a prior probability of 0.001 to assess

whether the positive associations were noteworthy or not. Finally, publication bias was confirmed by Begg's funnel plot [38] and Egger's test [39]. All statistical analyses were performed using Stata version 12.0 (Stata Corporation, College Station, TX, USA).

## Results

### Study characteristics

Fig 1 lists a flow diagram for identifying and including studies. Overall, a total of 108 studies were involved in the present study. Then, 7 studies were excluded because their data overlapped with another 7 studies. Finally, 75 articles were eligible in this meta-analysis. S2 Table list the main characteristics of these studies. There were 63 case–control studies from 56 publications on *CYP1A1* T3801C polymorphism (including 20,825 BC cases and 25,495 controls and 51 case–control studies from 46 publications on *CYP1A1* A2455G polymorphism (including 20,124 BC cases and 29,183 controls). In addition, ten and twelve studies were performed to analyze *CYP1A1* T3801C and A2455G polymorphisms in premenopausal women, and thirteen and seventeen studies were conducted to analyze *CYP1A1* T3801C and A2455G polymorphisms in postmenopausal women, respectively, as shown in S3 Table.

### Quantitative synthesis

Table 2 lists the results of association between the *CYP1A1* T3801C polymorphism and risk of BC. The *CYP1A1* T3801C polymorphism was associated with BC risk in overall population (CC *vs*. TT: OR = 1.34, 95% CI = 1.11–1.62; CC *vs*. TT + TC: OR = 1.27, 95% CI = 1.08–1.50; TC + CC *vs*. TT: OR = 1.11, 95% CI = 1.02–1.22). In subgroup analyses by ethnicity and menopausal status, a significantly increased BC risk was observed in Asians (CC *vs*. TT: OR = 1.27,

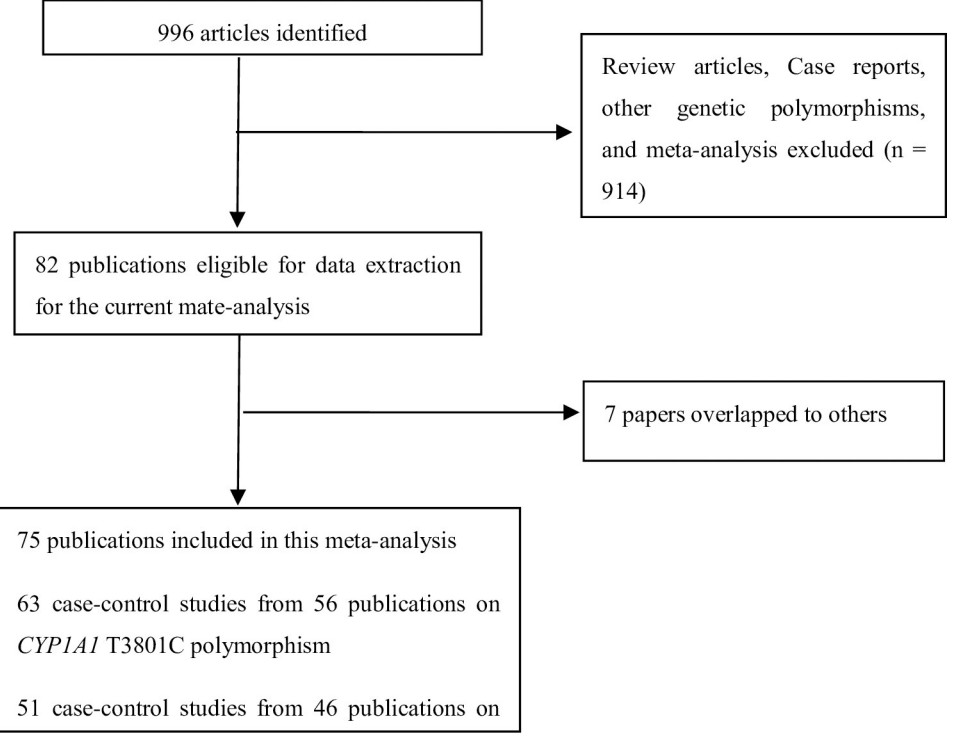

**Fig 1. Flow diagram for identifying and including studies in the current meta-analysis.**

**Table 2. Meta-analysis of the association of the *CYP1A1* T3801C polymorphism with risk of BC.**

| Variable | n (Cases/ Controls) | CC vs. TT | | | TC vs. TT | | | CC vs. TT+ TC | | | TC + CC vs.TT | | | C vs T | | |
|---|---|---|---|---|---|---|---|---|---|---|---|---|---|---|---|---|
| | | OR (95% CI) | $P_h/I^2$ (%) | BFDP | OR (95% CI) | $P_h/I^2$ (%) | BFDP | OR (95% CI) | $P_h/I^2$ (%) | BFDP | OR (95% CI) | $P_h/I^2$ (%) | BFDP | OR (95% CI) | $P_h/I^2$ (%) | BFDP |
| Overall | 63 (20825/ 25495) | **1.34 (1.11– 1.62)** | <0.001/ 74.8 | 0.986 | 1.07 (0.99– 1.17)* | <0.001/ 61.8 | – | **1.27 (1.08– 1.50)** | <0.001/ 69.6 | 0.993 | **1.11 (1.02– 1.22)** | <0.001/ 73.4 | 0.999 | – | <0.001/ 83.0 | – |
| Ethnicity | | | | | | | | | | | | | | | | |
| African | 6 (1231/ 1275) | 1.01 (0.58– 1.76) | 0.061/ 52.7 | – | 1.01 (0.82– 1.24) | 0.257/ 23.5 | – | 0.91 (0.54– 1.32) | 0.103/ 45.5 | – | 1.01 (0.81– 1.26) | 0.135/ 40.5 | – | 1.00 (0.81– 1.25) | 0.051/ 54.6 | – |
| Asian | 23 (6084/ 6529) | **1.27 (1.01– 1.59)** | <0.001/ 71.0 | 0.998 | 1.06 (0.93– 1.22) | 0.001/ 57.3 | – | 1.20 (0.99– 1.44) | <0.001/ 62.9 | – | 1.09 (0.94– 1.26) | <0.001/ 70.9 | – | – | <0.001/ 76.7 | – |
| Caucasian | 17 (7552/ 11364) | – | <0.001/ 77.3 | – | 1.09 (0.92– 1.28) | 0.001/ 61.4 | – | – | <0.001/ 75.2 | – | – | <0.001/ 75.3 | – | – | <0.001/ 86.9 | – |
| Indian | 5 (1009/ 944) | **2.68 (1.31– 5.51)** | 0.006/ 72.2 | 0.993 | – | <0.001/ 85.8 | – | **2.87 (2.02– 3.98)** | 0.100/ 48.5 | <0.001 | – | <0.001/ 88.5 | – | – | <0.001/ 89.1 | – |
| Menopausal status | | | | | | | | | | | | | | | | |
| Premenopausal | 10 (1605/ 1697) | 0.98 (0.74– 1.32) | 0.615/ 0.0 | – | 0.89 (0.65– 1.12) | 0.750/ 0.0 | – | 1.01 (0.78– 1.34) | 0.622/ 0.0 | – | 1.11 (0.82– 1.49) | 0.003/ 63.8 | – | 0.95 (0.82– 1.16) | 0.574/ 0.0 | – |
| Postmenopausal | 13 (5272/ 7946) | 1.23 (0.78– 1.95) | 0.027/ 55.8 | – | 0.98 (0.89– 1.08) | 0.545/ 0.0 | – | 1.23 (0.81– 1.87) | 0.043/ 51.7 | – | 1.06 (0.91– 1.23) | 0.070/ 39.6 | – | 1.06 (0.91– 1.23) | 0.072/ 46.2 | – |
| Sensitivity analysis (Only studies with high quality, matching, HWE, and genotyping examination done bindly or quality control) | | | | | | | | | | | | | | | | |
| Overall | 8 (6655/ 9181) | 1.02 (0.89– 1.21) | 0.132/ 35.8 | – | 1.00 (0.93– 1.09) | 0.234/ 23.6 | – | 1.04 (0.90– 1.20) | 0.375/ 7.2 | – | 1.00 (0.93– 1.08) | 0.108/ 39.1 | – | 1.03 (0.94– 1.13)* | 0.093/ 41.2 | – |
| Ethnicity | | | | | | | | | | | | | | | | |
| African | 1 (194/ 189) | 0.65 (0.27– 1.57) | – | – | 0.95 (0.62– 1.45) | – | – | 0.66 (0.28– 1.58) | – | – | 0.90 (0.60– 1.35) | – | – | 0.87 (0.62– 1.22) | – | – |
| Asian | 4 (2200/ 2403) | 1.19 (0.89– 1.59)* | 0.078/ 56.0 | – | 1.07 (0.84– 1.37)* | 0.025/ 67.9 | – | 1.09 (0.93– 1.28) | 0.360/ 6.7 | – | 1.11 (0.86– 1.42)* | 0.011/ 73.2 | – | 1.09 (0.92– 1.30)* | 0.016/ 70.8 | – |
| Caucasian | 3 (3872/ 6200) | 0.76 (0.48– 1.20) | 0.287/ 19.8 | – | 1.00 (0.90– 1.11) | 0.676/ 0.0 | – | 0.78 (0.50– 1.20) | 0.282/ 21.0 | – | 0.98 (0.88– 1.10) | 0.596/ 0.0 | – | 0.97 (0.88– 1.07) | 0.498/ 0.0 | – |
| Indian | – | – | – | – | – | – | – | – | – | – | – | – | – | – | – | – |
| Menopausal status | | | | | | | | | | | | | | | | |
| Premenopausal | 2 (814/ 827) | 0.94 (0.68– 1.32) | 0.186/ 42.7 | – | 0.96 (0.78– 1.19) | 0.492/ 0.0 | – | 0.97 (0.73– 1.27) | 0.227/ 31.4 | – | 0.92 (0.74– 1.21) | 0.281/ 14.1 | – | 0.89 (0.62– 1.26) | 0.149/ 51.9 | – |
| Postmenopausal | 3 (3622/ 6014) | 0.77 (0.56– 1.08) | 0.488/ 0.0 | – | 0.98 (0.88– 1.09) | 0.937/ 0.0 | – | 0.79 (0.58– 1.09) | 0.464/ 0.0 | – | 0.96 (0.87– 1.07) | 0.910/ 0.0 | – | 0.95 (0.86– 1.04) | 0.896/ 0.0 | – |

*a random-effects model was used; BC: breast cancer; HWE: Hardy-Weinberg equilibrium.

95% CI: 1.01–1.59) and Indians (CC *vs*. TT: OR = 2.68, 95% CI: 1.31–5.51; CC *vs*. TT + TC: OR = 2.87, 95% CI: 2.02–3.98). However, after using BFDP correction, associations remained significant only in Indians (CC *vs*. TT + TC: BFDP < 0.001).

**Table 3. Meta-analysis of the association of the *CYP1A1* A2455G polymorphism with risk of BC.**

| Variable | n (Cases/Controls) | GG vs. AA | | | AG vs. AA | | | GG vs. AA+ AG | | | AG + GG vs. AA | | | G vs A | | |
|---|---|---|---|---|---|---|---|---|---|---|---|---|---|---|---|---|
| | | OR (95% CI) | $P_h/I^2$ (%) | BFDP | OR (95% CI) | $P_h/I^2$ (%) | BFDP | OR (95% CI) | $P_h/I^2$ (%) | BFDP | OR (95% CI) | $P_h/I^2$ (%) | BFDP | OR (95% CI) | $P_h/I^2$ (%) | BFDP |
| Overall | 51 (20124/ 29183) | **1.39 (1.07– 1.82)** | <0.001/ 60.4 | 0.996 | 1.04 (0.94– 1.14) | <0.001/ 50.5 | – | **1.32 (1.04– 1.67)** | <0.001/ 55.4 | 0.960 | 1.08 (0.98– 1.20) | <0.001/ 63.5 | – | 1.10 (0.99– 1.23) | <0.001/ 72.8 | – |
| Ethnicity | | | | | | | | | | | | | | | | |
| African | 4 (829/ 872) | 0.97 (0.22– 4.26) | 0.406/ 0.0 | – | 0.91 (0.61– 1.36) | 0.516/ 0.0 | – | 0.98 (0.22– 4.31) | 0.397/ 0.0 | – | 0.91 (0.62– 1.35) | 0.619/ 0.0 | – | 0.92 (0.63– 1.34) | 0.718/ 0.0 | – |
| Asian | 11 (3760/ 4342) | 0.91 (0.78– 1.14) | 0.414/ 3.0 | – | 1.00 (0.91– 1.11) | 0.653/ 0.0 | – | 0.94 (0.78– 1.13) | 0.431/ 1.1 | – | 0.97 (0.89– 1.06) | 0.459/ 0.0 | – | 0.99 (0.91– 1.06) | 0.524/ 0.0 | – |
| Caucasian | 22 (11037/ 19156) | 1.88 (0.98– 3.59) | 0.022/ 46.4 | – | 0.98 (0.84– 1.15) | 0.020/ 43.7 | – | 1.73 (0.95– 3.14)* | 0.033/ 43.4 | – | 1.09 (0.90– 1.32)* | <0.001/ 68.3 | – | – | <0.001/ 78.8 | – |
| Indian | 5 (897/ 826) | **4.06 (1.09– 15.11)** | 0.012/ 68.7 | 0.998 | – | <0.001/ 82.8 | – | **3.59 (1.09– 11.80)*** | 0.031/ 62.4 | 0.973 | – | <0.001/ 86.7 | – | – | <0.001/ 86.7 | – |
| Menopausal status | | | | | | | | | | | | | | | | |
| Premenopausal | 12 (1497/ 1692) | 1.18 (0.57– 2.44) | 0.001/ 70.6 | – | 1.03 (0.86– 1.24) | 0.504/ 0.0 | – | 1.12 (0.58– 2.17) | 0.002/ 68.7 | – | 1.21 (0.92– 1.60) | 0.017/ 52.5 | – | 1.08 (0.82– 1.43) | 0.001/ 71.1 | – |
| Postmenopausal | 17 (6113/ 8965) | 1.32 (0.82– 2.14) | 0.099/ 38.8 | – | 1.06 (0.92– 1.23) | 0.158/ 31.3 | – | 1.10 (0.82– 1.54) | 0.311/ 14.4 | – | **1.27 (1.07– 1.50)** | 0.023/ 45.1 | 0.993 | 1.18 (0.99– 1.40) | 0.015/ 56.2 | – |
| Sensitivity analysis (Only studies with high quality, matching, HWE, and genotyping examination done bindly or quality control) | | | | | | | | | | | | | | | | |
| Overall | 7 (7260/ 9798) | 0.94 (0.72– 1.26) | 0.160/ 32.3 | – | 0.96 (0.85– 1.08) | 0.321/ 13.6 | – | 0.96 (0.72– 1.25) | 0.162/ 32.0 | – | 0.93 (0.86– 1.15) | 0.208/ 26.6 | – | 0.99 (0.88– 1.11) | 0.085/ 42.3 | – |
| Ethnicity | | | | | | | | | | | | | | | | |
| African | – | – | – | – | – | – | – | – | – | – | – | – | – | – | – | – |
| Asian | 3 (2010/ 2093) | 0.98 (0.60– 1.57) | 0.094/ 53.1 | – | 1.02 (0.90– 1.16) | 0.977/ 0.0 | – | 0.97 (0.60– 1.56) | 0.086/ 54.5 | – | 1.01 (0.89– 1.14) | 0.885/ 0.0 | – | 0.99 (0.90– 1.10) | 0.488/ 0.0 | – |
| Caucasian | 4 (4863/ 7316) | 0.86 (0.42– 1.97) | 0.260/ 25.2 | – | 0.90 (0.69– 1.16) | 0.078/ 56.0 | – | 0.82 (0.38– 1.98) | 0.277/ 22.3 | – | 0.90 (0.69– 1.18) | 0.057/ 60.1 | – | 0.91 (0.70– 1.19) | 0.042/ 63.5 | – |
| Indian | – | – | – | – | – | – | – | – | – | – | – | – | – | – | – | – |
| Menopausal status | | | | | | | | | | | | | | | | |
| Premenopausal | 1 (367/ 421) | 0.65 (0.34– 1.26) | – | – | 1.01 (0.75– 1.36) | – | – | 0.65 (0.34– 1.24) | – | – | 0.96 (0.72– 1.27) | – | – | 0.92 (0.73– 1.16) | – | – |
| Postmenopausal | 4 (4234/ 6646) | 1.02 (0.56– 1.52) | 0.125/ 47.7 | – | 1.01 (0.89– 1.15) | 0.588/ 0.0 | – | 1.01 (0.72– 1.63) | 0.122/ 48.2 | – | 1.01 (0.89– 1.14) | 0.392/ 0.0 | – | 1.02 (0.86– 1.18) | 0.176/ 39.3 | – |

BC: breast cancer; HWE: Hardy-Weinberg equilibrium.

**Table 3** shows the results of association between the *CYP1A1* A2455G polymorphism and risk of BC. The *CYP1A1* A2455G polymorphism was also associated with BC risk in the overall population (GG *vs.* AA: OR = 1.39, 95% CI = 1.07–1.82; GG *vs.* AA + AG: OR = 1.32, 95% CI = 1.04–1.67). In subgroup analyses by ethnicity and menopausal status, a statistically significant increased BC risk was yielded in Indians (GG *vs.* AA: OR = 4.06, 95% CI: 1.09–15.11; GG

*vs*. AA + AG: OR = 3.59, 95% CI: 1.09–11.80) and postmenopausal women (OR = 1.27, 95% CI: 1.07–1.50 for GG *vs*. AA + AG) for the *CYP1A1* A2455G polymorphism. However, after using BFDP correction, no significant associations were found in overall, Indians, and post-menopausal women.

## Heterogeneity and sensitivity analyses

Significant heterogeneity was observed in this study. Then, a meta-regression analysis was conducted to explore the source of heterogeneity by ethnicity, sample size, source of controls, type of controls, matching, HWE, and quality score. source of heterogeneity only be found in quality score (AG vs. AA: *P* = 0.031, G vs. A: *P* = 0.030) for the *CYP1A1* A2455G polymorphism.

Then, a sensitivity analysis was performed to assess the stability of results (as shown in Tables 2 and 3). The results did not change when a single study was deleted each time in the meta-analysis (Figures not shown). However, when we only included studies of high-quality, HWE, matching, and genotyping examination done blindly or with quality control, no significant association was observed between the *CYP1A1* T3801C and A2455G polymorphisms and risk of BC.

## Publication bias

Significant publication bias was confirmed by Begg's funnel plot and Egger's test (*CYP1A1* T3801C: TC + CC vs. TT: *P* = 0.036 and C vs. T: *P* = 0.033; *CYP1A1* A2455G: AG + GG vs. AA: *P* = 0.030). Figs 2–4 indicate that the results change (*CYP1A1* T3801C: TC + CC vs. TT: OR = 0.99, 95% CI: 0.90–1.10 and C vs. T: OR = 1.00, 95% CI: 0.90–1.11; *CYP1A1* A2455G: AG + GG vs. AA: OR = 0.97, 95% CI: 0.86–1.09) in overall analysis after using the nonparametric 'trim and fill' method.

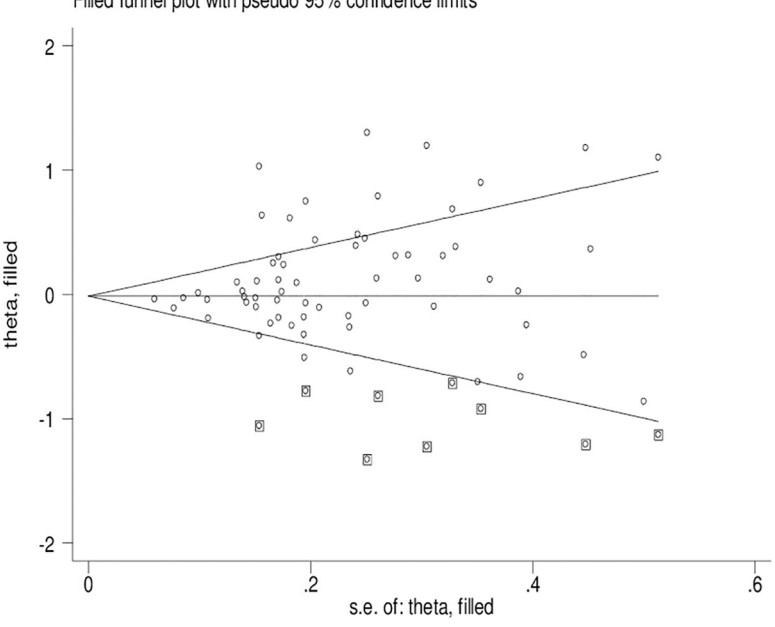

**Fig 2. The duval and tweedie nonparametric "trim and fill" method's funnel plot of the *CYP1A1* T3801C polymorphism (TC + CC vs. TT).**

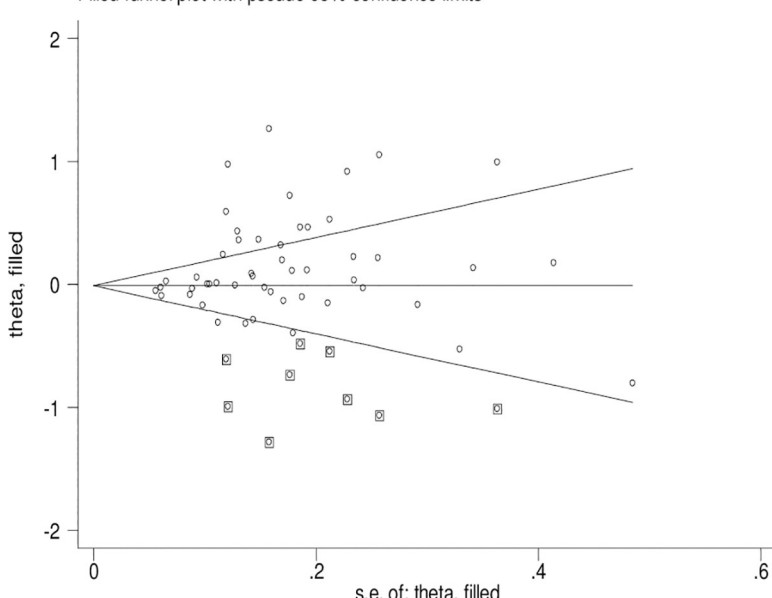

**Fig 3. The duval and tweedie nonparametric "trim and fill" method's funnel plot of the *CYP1A1* T3801C polymorphism (C vs. T).**

## Results of published meta-analyses

S4 Table shows the results of published meta-analyses for the *CYP1A1* T3801C and A2455G polymorphisms with BC risk in various different ethnic groups. Only one study [19] found that the *CYP1A1* T3801C polymorphism was significantly increased BC risk in Indians. Concerning the *CYP1A1* A2455G polymorphism, two studies [20, 21] observed a significantly

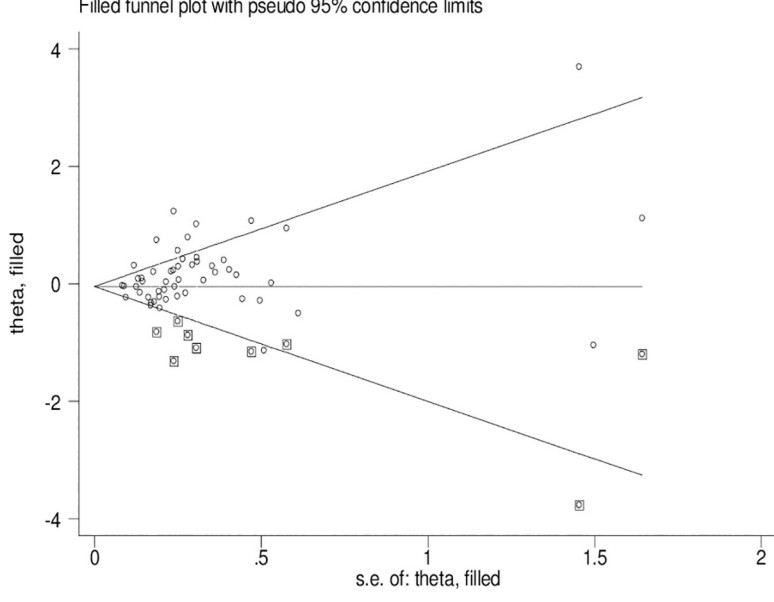

**Fig 4. The duval and tweedie nonparametric "trim and fill" method's funnel plot of the *CYP1A1* A3801G polymorphism (AG+GG vs. AA).**

increased BC risk in Caucasians and one study [22] found an obviously decreased BC risk in East Asians. However, when we used BFDP correction, only the *CYP1A1* T3801C polymorphism still be significant associated in Indians (CC vs. TT: BFDP < 0.001; TC + CC vs.TT: BFDP < 0.001).

## Discussion

Cytochrome P450s are enzymes which catalyze phase-I metabolism reactions. Cytochrome *P450 1A1* (*CYP1A1*) is one of the member of the *CYP* family and plays an important role in phase-I metabolism of polycyclic aromatic hydrocarbons as well as in estrogen metabolism. The dysfunction of *CYP1A1* can cause damages to DNA, lipids, and proteins, which further lead to carcinogenesis.

Overall, the *CYP1A1* T3801C polymorphism was significantly increased BC risk in overall analysis, especially in Asians and Indians; the *CYP1A1* A2455G polymorphism was associated with BC risk in overall analysis, Indians, and postmenopausal women. Published meta-analysis [19] found that the *CYP1A1* T3801C polymorphism was significantly increased BC risk in South Indians. Concerning the *CYP1A1* A2455G polymorphism, two meta-analyses [20, 21] observed a significantly increased BC risk in Caucasians and one study [22] found an obviously decreased BC risk in East Asians. As far as we know, meta-analyses of gene polymorphism and disease risk because they used several subgroups and genetic models at the expense of multiple comparisons, under these circumstances, the pooled *P*-value must be adjusted [40]. Wakefield et al. [37] proposed a precise Bayesian measure of false discovery in genetic epidemiology studies. Therefore, BFDP were considered to assess the significant associations in this study. When we used BFDP correction, associations remained significant only in Indians (CC *vs*. TT + TC: BFDP < 0.001) for *CYP1A1* T3801C polymorphism with BC risk. However, when we further performed sensitivity analysis, no significant association in overall analysis and any subgroups. Moreover, we found that all studies from Indians was low quality. Therefore, the results may be not credible. Further studies should be based on more high quality studies to confirm the association in Indians.

Obvious publication bias was observed by Begg's funnel plots and Egger's test between the *CYP1A1* T3801C polymorphism and BC risk in the current meta-analysis. Some small sample studies were easier to publish if there were positive results as they tend to obtain false-positive results because they may be not rigorous and are often of low-quality. In addition, random error and bias were common in small sample size, therefore, their conclusions may be unreliable on gene polymorphism with disease risk. Figs 2–4 also indicate that the asymmetry of the funnel plots were caused by some studies with low-quality small samples.

S4 Table shows the results of published meta-analyses for *CYP1A1* T3801C and A2455G polymorphisms with BC risk in various different ethnic groups (S1 Table). An significant inconsistency was observed in classification of ethnic groups among the published meta-analyses, especially for studies from USA, India, and Brazil (cells with red color in S1 Table). Moreover, we found that the published meta-analyses involved some repeat studies and many studies were also included. Furthermore, no studies adjusted positive results for multiple comparison using BFDP test.

Of these published meta-analyses, one involved studies only from African population [18], one from Chinese population [25], one from Indians [27], and the remaining examined all races [19–22, 24, 26]. Previous meta-analyses of maximum sample size was performed in 2014 for *CYP1A1* T3801C (47 studies 16,272 case and 20,930 controls) and A2455G (38 studies 15,969 case and 24,931 controls) with BC risk [19, 21]. The studies number and sample size of the present meta-analysis (63 studies including 20,825 BC cases and 25,495 controls for

T3801C and 51 studies including 20,124 BC cases and 29,183 controls) were larger than published meta-analyses. There were several deficiencies with the present study comparison. First, all previous meta-analyses [19–27] did not perform literature quality assessment. Second, all previous meta-analyses [19–26] did not adjusted positive results for multiple comparison excepting one study using FDR method [27]. Third, several published meta-analysis did not perform the sensitivity analysis. Moreover, previous meta-analyses included incomplete studies and some repeat studies did not be excluded (S1 and S4 Tables). Finally, An obvious inconsistency was found in classification of ethnic groups between these published meta-analyses, especially for studies from USA, India, Brazil, and so on (cells with blue color in S1 Table). Hence, we performed an updated meta-analysis to further explore the *CYP1A1* T3801C and A2455G polymorphism with BC risk. In the current meta-analysis, a larger sample size was collected. In addition, we evaluated quality assessment of the eligible studies. Moreover, we applied meta-regression analysis to investigate the source of heterogeneity. Further, we performed a sensitivity analysis, especially we used a data set only including studies of high-quality, matching, HWE, and in which genotyping was performed blindly or with quality control (this was an attempt to avoid random errors and confounding bias that sometimes distorted the results of molecular epidemiological studies). Finally, we used BFDP method to assess the significant associations.

Despite all our efforts to improve our research. However, this study still exists several limitations. First, only published articles were included, so publication bias may be unavoidable. Second, some subgroup analyses included less studies, for instance, there were only five studies on the *CYP1A1* T3801C polymorphism with BC risk in Indians and four studies on the *CYP1A1* A2455G polymorphism with BC risk in Africans. Third, data were not stratified by age, family history, smoking status, and other environmental factors. Hence, a more precise analysis should be performed when enough data was available in future.

## Conclusions

In summary, this meta-analysis strongly indicates that there is no significantly associated between the *CYP1A1* T3801C and A2455G polymorphisms and BC risk. The increased BC risk may most likely on account of false-positive results. Significant association should be interpreted with caution and it is essential that future analysis be based on sample sizes well-powered to identify these variants having modest effects on BC risk, especially the combined effects, such as gene-gene and gene-environmental.

## Supporting information

**S1 Table. Included studies of the *CYP1A1* polymorphisms in BC within the meta-analyses.**
(PDF)

**S2 Table. Genotype distribution of the *CYP1A1* polymorphisms in the included studies of BC.**
(PDF)

**S3 Table. Genotype frequencies of the *CYP1A1* polymorphisms between and breast cancer and control groups by menopausal status.**
(PDF)

**S4 Table. Results of previous meta-analyses between CYP1A1 T3801C and A2455G polymorphisms with BC risk.**
(PDF)

**S1 Appendix. References.**
(DOCX)

**S1 File. PRISMA checklist.**
(DOC)

**S2 File. Meta analysis on genetic association studies form.**
(DOCX)

## Author Contributions

**Data curation:** Chen Yang, Xiao-Feng He.

**Formal analysis:** Chen Yang, Xiao-Feng He.

**Investigation:** Chen Yang, Xiao-Feng He.

**Resources:** Xiao-Feng He.

**Supervision:** Xiao-Feng He.

**Writing – original draft:** Chen Yang.

**Writing – review & editing:** Xiao-Feng He.

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
