## [Decision Letter · Decision Letter 0]

31 Dec 2020

PONE-D-20-26207

Evaluation of association studies, a systematic review and meta-analysis of CYP1A1 T3801C and A2455G polymorphisms and risk of breast cancer

PLOS ONE

Dear Dr. He,

Thank you for submitting your manuscript to PLOS ONE. After careful consideration, we feel that it has merit but does not fully meet PLOS ONE’s publication criteria as it currently stands. Therefore, we invite you to submit a revised version of the manuscript that addresses the points raised during the review process.

The manuscript require English language editing

We look forward to receiving your revised manuscript.

Kind regards,

Shama Prasada Kabekkodu

Academic Editor

PLOS ONE

Journal Requirements:

2.We suggest you thoroughly copyedit your manuscript for language usage, spelling, and grammar. If you do not know anyone who can help you do this, you may wish to consider employing a professional scientific editing service.  

3.In your Data Availability statement, you have not specified where the minimal data set underlying the results described in your manuscript can be found. PLOS defines a study's minimal data set as the underlying data used to reach the conclusions drawn in the manuscript and any additional data required to replicate the reported study findings in their entirety. All PLOS journals require that the minimal data set be made fully available. For more information about our data policy, please see http://journals.plos.org/plosone/s/data-availability.

4.Thank you for stating the following financial disclosure:

 "No

Additional Editor Comments (if provided):

The manuscript require English language editing.

Reviewers' comments:

Reviewer's Responses to Questions

**Comments to the Author**

1. Is the manuscript technically sound, and do the data support the conclusions?

Reviewer #1: Yes

Reviewer #2: Yes

2. Has the statistical analysis been performed appropriately and rigorously? 

Reviewer #1: Yes

Reviewer #2: Yes

3. Have the authors made all data underlying the findings in their manuscript fully available?

Reviewer #1: Yes

Reviewer #2: Yes

4. Is the manuscript presented in an intelligible fashion and written in standard English?

Reviewer #1: Yes

Reviewer #2: Yes

5. Review Comments to the Author

Reviewer #1: In the article: “Evaluation of association studies, a systematic review and meta-analysis of CYP1A1 T3801C and A2455G polymorphisms and risk of breast cancer” the authors analyse two polymorphisms of CYP1A1 in different populations and discuss all previous studies on this subject.

1. In my opinion this article is well prepared and concise. As the Authors write: they “performed an updated systematic review and meta-analysis”.

I have only some comments to this article:

2. There is a lack of information in Abstract how many cancer patients and controls were evaluated?

"The studies number and sample size of the present meta-analysis (63 studies

including 20,825 BC cases and 25,495 controls for T3801C and 51 studies including 20,124 BC

cases and 29,183 controls)"

3. There is MTHFR in abstract and in Discussion instead of CYP1A (twice).

“When we used BFDP correction, associations remained significant only in Indians (CC vs. TT + TC: BFDP< 0.001) for MTHFR T3801C polymorphism with BC”

Reviewer #2: This article showed relevant findings regarding breast cancer genetic risk.

However I put forward some minor comments and suggestions, which needs to be addressed by the authors.

1. I thought the objective in the abstract can be better stated that would clarify the research rationale. In abstract, the concluding remarks states “This meta-analysis strongly indicates that there is no significantly associated between the CYP1A1 T3801C and A2455G polymorphisms and BC risk”; instead of significantly associated, it would be significant association.

2. In the Introduction section, the authors should discuss the need of performing the meta-analysis in the light of the GWAS findings and justify the need of performing the meta-analysis.

3. What is the basis of considering 12 as the cut-off score, as stated in the Data extraction and Quality assessment module?

4. I suggest that choosing the model based on the Q-test is ill-advised.

See, for example:

a. Hedges, L. V., &Vevea, J. L. (1998). Fixed- and random-effects models in meta-analysis. Psychological Methods, 3(4), 486-504.

b. Borenstein, M., Hedges, L. V., Higgins, J. P. T., & Rothstein, H. R. (2010). A basic introduction to fixed-effect and random-effects models for meta-analysis. Research Synthesis Methods, 1(2), 97-165.

c. "https://training.cochrane.org/ handbook/current/chapter-10"https://training.cochrane.org/handbook/current/chapter-10#section-10-10-4-1

The authors need to choose the model based on their hypothesis or present the results generated in both the models.

5. Recently published meta-analysis (https://doi.org/10.2217/fon-2020-0333) on the Indian subcontinent population showed similar findings on CYP1A1 T3801C along with an in silico analysis of to explain the biological role. This shows collinearity with the findings of this manuscript and it needs to be discussed.

6. How does the publication bias affect the findings of this study? The authors need to discuss in details.

7. The current version of the manuscript, however, needs improvement with respect to overall language and structuring.

6. PLOS authors have the option to publish the peer review history of their article (what does this mean?). If published, this will include your full peer review and any attached files.

Reviewer #1: **Yes: **Izabela Laczmanska

Reviewer #2: No

---

## [Author Response · Author response to Decision Letter 0]

27 Feb 2021

Thank you very much for sending us the valuable comments, which is very helpful to improve the quality of our study. A thoroughly revised vision of our artwork has been made. The comments have been addressed and itemized as follows:

1.Please ensure that your manuscript meets PLOS ONE's style requirements, including those for file naming.

We ensured that our manuscript met Plos one's style requirements including file naming.

2.We suggest you thoroughly copy edit your manuscript for language usage, spelling, and grammar. 

 We have revised the grammatical errors and improved the overall readability of the text. 

3.In your Data Availability statement, you have not specified where the minimal data set underlying the results described in your manuscript can be found. PLOS defines a study's minimal data set as the underlying data used to reach the conclusions drawn in the manuscript and any additional data required to replicate the reported study findings in their entirety. All PLOS journals require that the minimal data set be made fully available. For more information about our data policy, please see http://journals.plos.org/plosone/s/data-availability.

All relevant data are within the paper and its Supporting Information files.

4.Thank you for stating the following financial disclosure:

The funders had no role in study design, data collection and analysis, decision to publish, or preparation of the manuscript." At this time, please address the following queries:

1.Please clarify the sources of funding (financial or material support) for your study. List the grants or organizations that supported your study, including funding received from your institution. 2.State what role the funders took in the study. If the funders had no role in your study, please state: “The funders had no role in study design, data collection and analysis, decision to publish, or preparation of the manuscript.”

3.If any authors received a salary from any of your funders, please state which authors and which funders.

4.If you did not receive any funding for this study, please state: “The authors received no specific funding for this work.”

 We have state: “The authors received no specific funding for this work” in this manuscript.

 Additional Editor Comments (if provided):

The manuscript require English language editing.

We have revised the grammatical errors and improved the overall readability of the text. 

Reviewers' comments:

Reviewer's Responses to Questions

Comments to the Author

1. Is the manuscript technically sound, and do the data support the conclusions?

Reviewer #1: Yes

Reviewer #2: Yes

2. Has the statistical analysis been performed appropriately and rigorously?

Reviewer #1: Yes

Reviewer #2: Yes

3. Have the authors made all data underlying the findings in their manuscript fully available?

Reviewer #1: Yes

Reviewer #2: Yes

4. Is the manuscript presented in an intelligible fashion and written in standard English?

Reviewer #1: Yes

Reviewer #2: Yes

Reviewer #1: In the article: “Evaluation of association studies, a systematic review and meta-analysis of CYP1A1 T3801C and A2455G polymorphisms and risk of breast cancer” the authors analyse two polymorphisms of CYP1A1 in different populations and discuss all previous studies on this subject. In my opinion this article is well prepared and concise. As the Authors write: they “performed an updated systematic review and meta-analysis”.

I have only some comments to this article:

1. There is a lack of information in Abstract how many cancer patients and controls were evaluated? “The studies number and sample size of the present meta-analysis (63 studies

including 20,825 BC cases and 25,495 controls for T3801C and 51 studies including 20,124 BC cases and 29,183 controls)”.

We have added the studies number and sample size of the present meta-analysis.

2. There is MTHFR in abstract and in Discussion instead of CYP1A (twice). “When we used BFDP correction, associations remained significant only in Indians (CC vs. TT + TC: BFDP< 0.001) for MTHFR T3801C polymorphism with BC”.

We have changed MTHFR to CYP1A1 in abstract and Discussion sections!

Reviewer #2: This article showed relevant findings regarding breast cancer genetic risk.

However I put forward some minor comments and suggestions, which needs to be addressed by the authors.

1. I thought the objective in the abstract can be better stated that would clarify the research rationale. In abstract, the concluding remarks states “This meta-analysis strongly indicates that there is no significantly associated between the CYP1A1 T3801C and A2455G polymorphisms and BC risk”; instead of significantly associated, it would be significant association.

We have revised the abstract by your comment. 

2. In the Introduction section, the authors should discuss the need of performing the meta-analysis in the light of the GWAS findings and justify the need of performing the meta-analysis.

We have discussed the GWAS in the Introduction section.

3. What is the basis of considering 12 as the cut-off score, as stated in the Data extraction and Quality assessment module?

Studies with scores 0-7, 8-13, or 14-20 were of low, moderate, or high-quality by two previously published meta-analyses [39-40], respectively. At present, the sensitivity analysis including high-quality studies are > 14.

4. I suggest that choosing the model based on the Q-test is ill-advised. See, for example: Hedges, L. V., &Vevea, J. L. (1998). Fixed- and random-effects models in meta-analysis. Psychological Methods, 3(4), 486-504. Borenstein, M., Hedges, L. V., Higgins, J. P. T., & Rothstein, H. R. (2010). A basic introduction to fixed-effect and random-effects models for meta-analysis. Research Synthesis Methods, 1(2), 97-165. "https://training.cochrane.org/ handbook/current/chapter-10"https://training.cochrane.org/handbook/current/chapter-10#section-10-10-4-1. The authors need to choose the model based on their hypothesis or present the results generated in both the models.

For each genetic model contrast, summary ORs were calculated using random-effects model [26, 27]. The random-effects model was applied by the following two main reasons: (1) because the Q test is characterized by low statistical power for between-study heterogeneity, which is especially relevant when few studies are available; (2) Usually, the random-effects model is a more conservative choice when heterogeneity is present, whereas it reduces to the fixed effect model when heterogeneity is absent. Hence, recalculated the pooled studies only using the random-effects model.

5. Recently published meta-analysis (https://doi.org/10.2217/fon-2020-0333) on the Indian subcontinent population showed similar findings on CYP1A1 T3801C along with an in silico analysis of to explain the biological role. This shows collinearity with the findings of this manuscript and it needs to be discussed.

We have discussed the recently published meta-analysis on the CYP1A1 T3801C polymorphism with BC risn in Indian population.

6. How does the publication bias affect the findings of this study? The authors need to discuss in details.

Publication bias was found between the CYP1A1 T3801C polymorphism and BC risk in the current meta-analysis. Some small sample studies were easier to publish if there were positive results as they tend to obtain false-positive results because they may be not rigorous and are often of low-quality. Figs 2-4 also indicate that the asymmetry of the funnel plots were caused by some studies with low-quality small samples.

7.The current version of the manuscript, however, needs improvement with respect to overall language and structuring.

We have revised the language and structuring.

---

## [Decision Letter · Decision Letter 1]

23 Mar 2021

Evaluation of association studies and a systematic review and meta-analysis of CYP1A1 T3801C and A2455G polymorphisms in breast cancer risk

PONE-D-20-26207R1

Dear Dr. He,

We’re pleased to inform you that your manuscript has been judged scientifically suitable for publication and will be formally accepted for publication once it meets all outstanding technical requirements.

Kind regards,

Shama Prasada Kabekkodu

Academic Editor

PLOS ONE

Additional Editor Comments (optional):

Authors have satisfactorily addressed the reviewers query.

Reviewers' comments:

Reviewer's Responses to Questions

**Comments to the Author**

1. If the authors have adequately addressed your comments raised in a previous round of review and you feel that this manuscript is now acceptable for publication, you may indicate that here to bypass the “Comments to the Author” section, enter your conflict of interest statement in the “Confidential to Editor” section, and submit your "Accept" recommendation.

Reviewer #2: All comments have been addressed

2. Is the manuscript technically sound, and do the data support the conclusions?

Reviewer #2: Yes

3. Has the statistical analysis been performed appropriately and rigorously? 

Reviewer #2: Yes

4. Have the authors made all data underlying the findings in their manuscript fully available?

Reviewer #2: Yes

5. Is the manuscript presented in an intelligible fashion and written in standard English?

Reviewer #2: No

6. Review Comments to the Author

Reviewer #2: All queries addressed to satisfaction both scientifically and technically. Overall efforts appreciated

7. PLOS authors have the option to publish the peer review history of their article (what does this mean?). If published, this will include your full peer review and any attached files.

Reviewer #2: **Yes: **Mainak Sengupta

---

## [Editor Report · Acceptance letter]

7 Apr 2021

PONE-D-20-26207R1 

Evaluation of association studies and a systematic review and meta-analysis of *CYP1A1* T3801C and A2455G polymorphisms in breast cancer risk 

Dear Dr. He:

I'm pleased to inform you that your manuscript has been deemed suitable for publication in PLOS ONE. Congratulations! Your manuscript is now with our production department. 

Kind regards, 

on behalf of

Dr. Shama Prasada Kabekkodu 

Academic Editor

PLOS ONE